# Chitosan Reinforced with Kenaf Nanocrystalline Cellulose as an Effective Carrier for the Delivery of Platelet Lysate in the Acceleration of Wound Healing

**DOI:** 10.3390/polym13244392

**Published:** 2021-12-15

**Authors:** Payal Bhatnagar, Jia Xian Law, Shiow-Fern Ng

**Affiliations:** 1Centre for Drug Delivery Technology, Faculty of Pharmacy, Universiti Kebangsaan Malaysia, Jalan Raja Muda Abdul Aziz, Kuala Lumpur 50300, Malaysia; payalbhatnagar2020@gmail.com; 2Centre for Tissue Engineering & Regenerative Medicine, 12th Floor, Clinical Block, UKM Medical Centre, Jalan Yaa’cob Latif, Cheras, Kuala Lumpur 56000, Malaysia

**Keywords:** chitosan, kenaf, nanocrystalline cellulose, platelet lysate, wound healing

## Abstract

The clinical use of platelet lysate (PL) in the treatment of wounds is limited by its rapid degradation by proteases at the tissue site. This research aims to develop a chitosan (CS) and kenaf nanocrystalline cellulose (NCC) hydrogel composite, which intend to stabilize PL and control its release onto the wound site for prolonged action. NCC was synthesized from raw kenaf bast fibers and incorporated into the CS hydrogel. The physicochemical properties, in vitro cytocompatibility, cell proliferation, wound scratch assay, PL release, and CS stabilizing effect of the hydrogel composites were analyzed. The study of swelling ratio (>1000%) and moisture loss (60–90%) showed the excellent water retention capacity of the CS-NCC-PL hydrogels as compared with the commercial product. In vitro release PL study (flux = 0.165 mg/cm^2^/h) indicated that NCC act as a nanofiller and provided the sustained release of PL compared with the CS hydrogel alone. The CS also showed the protective effect of growth factor (GF) present in PL, thereby promoting fast wound healing via the formulation. The CS-NCC hydrogels also augmented fibroblast proliferation in vitro and enhanced wound closures over 72 h. This study provides a new insight on CS with renewable source kenaf NCC as a nanofiller as a potential autologous PL wound therapy.

## 1. Introduction

In the last decade, biopolymers have attracted substantial attention mainly because of their renewability, biocompatibility, biodegradability, and abundance. Nanocellulose, a biopolymer derived from plants, is a promising biomaterial in skin grafts, implants, tissue engineering, and wound healing. Nanocellulose are classified into nanocrystalline cellulose (NCC) and cellulose nanofibrils (CNF). NCC consists of amorphous and highly ordered crystalline cellulose structures. NCC exhibits overwhelming performance as a nanomaterial because of its dimensions 5–100 nm in diameter with lengths up to hundreds of nanometers (100–600 nm). NCC can be obtained from inexpensive biomass renewable resources, such as wood, shrubs, and herbs. Plant-derived NCC is highly beneficial for wound dressing applications because of its high capability to absorb liquids and form translucent films [1]. These properties are crucial for non-healing and chronic wounds, where exudates need to be managed adequately. In addition, the translucency of NCC allows the wound development to be evaluated without needing to remove the dressing. Agricultural waste is a valuable source of advanced biomaterials. Nanocellulose may be obtained from agro-waste and further used as reinforcement in biopolymers. Kenaf (*Hibiscus cannabinus* L.) is an important industrial crop with high economic and ecological importance in many tropical countries, including Malaysia. Kenaf plant components that are of industrial importance include bast and core fibers. Bast fibers are isolated from the outer layer of plant fibers (30% of dry weight), whereas the core refers to the inner fiber layer (70% of dry weight). Bast fibers have been extensively investigated for their excellent mechanical properties because of their high length-to-diameter ratio and superior crystallinity, allowing these fibers to be applied as a reinforcing agent with other natural or synthetic polymers for biomedical applications [2]. NCC isolated from kenaf waste has been extended to various scientific research for their special characteristics, such as sustainability, affordability, and well-established mechanical, thermal, and electrical properties, which attracted scientists to develop new promising materials with unique values [3]. Currently, the use of kenaf-derived nanocelluloses is limited to the packaging and paper industries. The use of kenaf-based NCC as wound dressing has yet to be researched and explored.

Platelet lysate (PL) derived from platelet-rich plasma (PRP) is an autologous source of therapeutic proteins that is extensively researched in hard-to-heal wounds and tissue regeneration [4,5,6]. PL is a rich source of growth factors (GFs), including epidermal growth factor (EGF), insulin-like growth factor (IGF), transforming growth factor beta (TGF-β), platelet-derived growth factor (PDGF), which are crucial in regulating cell proliferation, migration, and differentiation by binding to specific transmembrane receptors on target cells. EGF secreted during the hemostasis of inflammation by platelets exhibits chemotactic effect on keratinocytes by promoting re-epithelialization [7,8]. TGF-β has three isoforms (i.e., TGF-β; 1–3). TGF-β1 is highly produced during wound healing by platelets, neutrophils, macrophages, and fibroblasts and is responsible for the synthesis of extracellular matrix components. IGF is discharged by platelets at the beginning of hemostasis. It entices leukocytes and participates in cell inflammation and proliferation. Thus, IGF serves a regulatory function in fibroblast proliferation in tissues. The PL serves as the cellular induction of normal wound healing responses and PL therapy has clinical significance regardless of the wound etiology. Hence, PL applications have gained considerable popularity in tissue regeneration and wound healing. GFs are released quickly from PL; thus, their activity and clinical efficacy are easily lost [9]. As a result, hydrogels or carriers suitable for PL are highly sought after by researchers.

Chitosan (CS) is a polysaccharide which chemically exist as (1→4)-2 amino-2-deoxy-β-D-glucan; originated from chitin, a key constituent of crustacean outer skeletons commonly used as an antimicrobial agent for preventing and treating infections owing to its natural antimicrobial property [10]. CS shows unique hemostatic properties and promotes the infiltration and migration of neutrophils and macrophages in early stages of wound healing, making it a suitable biopolymer for wound dressing [11]. CS exists in the form of 3D hydrogel polymer networks that can absorb and retain a large quantity of moisture on account of the abundance of hydrophilic groups [12]. CS dressings provoke minimal adverse reactions with little or almost absent fibrous encapsulation and provide protection against bacterial infections [13]. Recently, CS and modified CS have been deemed suitable for the delivery of PRP or PL to wounds [6,7,8,9,10,11,12,13,14,15]. Rossi et al. demonstrated via PDGF evaluation that CS dressings maintain the platelet GF in unaltered active form [14]. CS fibers also tightly bind with major plasma proteins and at specific platelet surface proteins. The combination of CS with PL shows the sustained release of GFs and increases glycoprotein IIIa expression in platelets [16].

Despite the significant therapeutic potential of PL in wound repair, PL generally suffers clinical limitations, such as short half-life, instability, degradation by protease at the tissue site, and toxicity at high systemic dose [17]. An efficient delivery system has been sought to stabilize PL for its sustained release in wound healing. The applicability of NCC as a potential cell culture scaffold has been previously reported because it provides the desired 3D environments for the growth and differentiation of skin cells [18,19]. A successful PL delivery system from hydrogels could be developed by mimicking tissue regeneration in terms of endogenous release profiles of PL. This research aimed to develop a CS hydrogel reinforced with NCC and loaded with PL as a controlled-release vehicle for wound healing. CS with a 2% *w*/*v* was employed as a PL stabilizing agent and cell growth-promoting polymer and added with NCC 0.4% *w*/*v* to provide a nanocellulose network that can facilitate the loading and delivery of PL in wound healing.

## 2. Materials and Methods

### 2.1. Materials

Kenaf raw bast fiber was provided by the National Kenaf and Tobacco Board (Lembaga Kenaf dan Tembakau Negara, LKTN), Malaysia. Low-molecular-weight (LMW, analytical grade) CS (molecular weight of 50–190 kDa, 75–85% degree of deacetylation) was purchased from Sigma Aldrich (Dublin, Ireland), glacial acetic acid (GAA; 100%) was purchased from R.M Chemicals Pvt. Ltd. (Chandigarh, India), PL, cell culture media, and primary skin cells were received from Centre For Tissue Engineering and Regenerative Medicine (CTERM), UKM. Enzyme-linked immunosorbent assay (ELISA) kit was obtained from Insphiro Technology (Selangor, Malaysia). Alamar Blue was obtained from Invitrogen (Waltham, MA, USA). Dulbecco’s phosphate-buffered saline (DPBS) and fetal bovine serum (FBS) were purchased from Biowest (Riverside, MO, USA). Collagenase type I was obtained from Worthington (Columbus, OH, USA), Dulbecco’s modified Eagle’s medium (F12: DMEM; 1:1), antibacterial–antimycotic, Glutamax, and 4-(2-hydroxyethyl)-1-piperazine ethane sulfonic acid (HEPES) were procured from Gibco (Grand Island, NY, USA). All chemicals, such as sodium hydroxide (NaOH), sodium chlorite (NaClO_2_), and sulfuric acid (H_2_SO_4_), were analytical grade (Dublin, Ireland).

### 2.2. Preparation of Nanocrystalline Cellulose from Kenaf Bast Fibers

NCC was extracted from raw kenaf bast fibers adapted from previous method with slight modifications [20]. Briefly, the retting of raw bast fibers was carried out by soaking it in distilled water overnight and filtered several times with subsequent drying at 60 °C in the oven. The dried fibers were ground and sieved to obtain finer fibers. Thereafter, pulverization was carried out by treating the fibers with 5% (*w*/*v*) NaOH at 80 °C for 2 h, and this step was carried out thrice to accomplish delignification. The fibers were then thoroughly washed and filtered using distilled water to remove residual chemicals. Subsequently, the alkali-treated fibers were bleached using a combination of acetic acid and sodium chlorite to remove residual lignin and hemicellulose. The bleached pulp was filtered and washed numerous times after a series of bleaching procedures until the pH reached 7. The bleaching treatment was repeated five times at 80 °C for 3 h to obtain white fibers and thereafter stored in water swollen state. Finally, acid hydrolysis was commenced by immersing 4% (*w*/*w*) bleached kenaf bast fibers into 65% (*v*/*v*) sulfuric acid at a temperature of 45 °C for 30 min. The fiber suspension was constantly homogenized using a magnetic stirrer. Then, the fiber suspension was diluted by adding distilled water and allowed to cool. Afterwards, the acid was removed by centrifuging the fiber suspension at 12,000× *g* rpm for 30 min, and this process was repeated five times for complete removal of residual acid. Then, the fiber pellet was dispersed in distilled water and poured into a dialysis bag (MWCO 11 KDa; cellulose acetate) under slow stirring until complete neutralization. Subsequently, the suspension of tiny fibers was ultrasonicated for 5 min at a frequency of 20 kHz with an amplitude of 80%. This process was carried out under an ice box to prevent overheating. The resulting thick suspension containing the NCC was stored at 4 °C and freeze-dried for further characterization and hydrogel preparation.

### 2.3. PL Processing and Quantification of GFs

The PL was processed aseptically by freeze-thawing for one cycle and subsequently centrifuged at 5000× *g* rpm at 4 °C for 20 min. The resulting pellet containing cell debris was discarded and the processed PL was kept at −80 °C with the addition of 40 IU of LMW heparin as an anticoagulant until further use.

The concentrations of key GFs (EGF and TGF) were determined using ELISA based on Sandwich-ELISA using 96-well plates in accordance with the manufacturer’s instructions. Optical density (OD) were measured under 450 nm wavelength using a microplate reader Bio-Rad (Berkeley, CA, USA).

### 2.4. Preparation of CS Hydrogel and CS-NCC Composite Hydrogels

The porous CS hydrogels were prepared as previously described method with slight modifications [21]. In brief, 2% (*w*/*v*) CS solution was prepared by dissolving fine CS powder in 1% GAA, and pH was adjusted to 7.0 using 1 M NaOH. Subsequently, the solution was stirred on a magnetic stirrer for 1 h at room temperature. Afterward, CS solution was transferred into 24-well tissue-culture plates and frozen at −20 °C for 24 h, followed by freeze drying (Labconco, Topeka, KS, USA, Model no.117; weight-24 kg) at −80 °C for 4 days to ensure complete drying.

CS hydrogels containing PL (CS-PL) were made by mixing 1 mL of PL/gram of CS hydrogel (2% *w*/*v*) on a magnetic stirrer for 30 min at 20 °C. Then, the mixture was transferred to a 24-well tissue-culture plate for lyophilization at −80 °C for 4 days.

As for the CS-NCC composite hydrogel, NCC was dispersed in deionized water (DI) to produce a 0.4% (*w*/*v*) concentration [22]. Then, the slurry of NCC was added to the 2% (*w*/*v*) solution of CS. The resulting dispersion was ultra-sonicated at an amplitude of 40% for 5 min to homogenize NCC in CS solution and then allowed to mix for 1 h on a magnetic stirrer in a closed reactor. Then, acetic acid (1%) was added to solubilize the CS and the mixture was mechanically stirred for the next 5 h to obtain a complete dissolution of CS. Consequently, hydrogels were formed through the neutralization of the viscous suspension of CS by pouring it on a petri plate containing 1 M NaOH for 1 h. The prepared hydrogels were washed with distilled water until a neutral pH was obtained. Finally, the composite hydrogel of CS-NCC without using any specific cross-linkers were prepared and subsequently freeze-dried for further characterization. The CS-NCC-PL gel was prepared by thorough mixing of the CS-NCC gel with PL (1 mL/g). The CS-NCC gel was stored in the refrigerator at 4 °C. Upon use, PL (stored at −80 °C) was thawed to room temperature prior and added into the CS-NCC gel via simple mixing. The unused CS-NCC-PL hydrogel was stored at −20 °C but thawed to room temperature before tests.

### 2.5. Characterization of NCC and Hydrogels

#### 2.5.1. Chemical Composition, Fiber Yield, and Zeta Potential

Chemical composition of NCC was determined in accordance with TAPPI standard methods T 222 (acid-insoluble lignin in wood and pulp) [23]. The fiber yield of NCC was calculated in terms of percentage (%) of the initial weight of bleached fibers after hydrolysis. The suspension of the fiber obtained after dialysis treatment was freeze-dried and compared with the initial weight of fiber. The final weight of NCC (*Mf*) and the initial weight of fiber (*Mi*) were measured to calculate the yield using Equation (1) [24]. Fiber yield was calculated using the following equation.
Yield (%) = (*Mf*/*Mi*) × 100(1)

The zeta-potential of NCC was determined using Zetasizer Nano-ZS (Malvern Instruments Ltd., Malvern, UK) to identify the electrical charges of NCC. The NCC samples were tested after acid hydrolysis of the fiber suspension.

#### 2.5.2. X-ray Diffraction Characterization (XRD)

XRD patterns of bleached and NCC from kenaf bast fibers were performed on X-ray diffractometer (Empyrean PANalytical, Marvin, UK) to examine the changes in crystallinity before and after chemical treatment. The diffraction intensity of Cu Kα radiation (λ = 0.1542 nm; 40 kV and 40 mA) was measured in a 2θ range between 5° and 70° with scan rate of 0.5° per min. Peak analysis was performed using Diffrac.EvaV4.0 software. Meanwhile, the crystallinity index of fibers was calculated as previously described method [25] using the following Equation (2).
CrI (%) = [I_002_ − I_amorph_/I_002_] × 100(2)
where CrI is the crystallinity index is the maximum peak intensity at the (002) plane (around 22.5° for native cellulose) and I_amorph_ is the minimum intensity of the amorphous portion taken at 2θ = 18°

#### 2.5.3. Scanning Electron Microscopy (SEM)

A SEM instrument (Carl Zeiss Merlin Compact-Germany, Oberkochen, Germany) was used to observe the surface morphology of NCC. The acceleration voltage was set up in the range of 5 to 20 kV, and dried samples were sputter-coated with gold to avoid the charging effect during SEM observations. The fiber diameter was measured by using Smart TIFF image viewer software.

#### 2.5.4. Fourier Transform Infrared Spectroscopy-Attenuated Total Reflectance (FTIR-ATR) Spectroscopy

FTIR-ATR spectroscopy (Spectrum 100; Perkin Elmer, Walthman, MA, USA) was performed on raw fibers, bleached fibers, NCC, CS, PL, CS-NCC, CS-PL, and CS-NCC-PL hydrogels.

Infrared spectra of CS hydrogel control, PL, NCC, CS hydrogel-reinforced PL, and NCC were determined between 4000 and 650 cm^−1^ using FTIR-ATR spectrometer (Spectrum 100; Perkin Elmer, Walthman, MA, USA). The spectra were acquired using 32 scans and a 4 cm^−1^ resolution.

#### 2.5.5. Swelling Behavior of Hydrogels

The swelling behavior of freeze-dried samples of CS, CS-NCC, CS-NCC-PL, CS-PL hydrogel, and Intrasite™ gel as positive control were tested in DI at 37 °C for 3, 6, and 24 h. The dried samples were weighed and placed in 20 mL of DI, and the hydrogels were allowed to reach their swelling equilibrium. The weights of the hydrogels (W2) were measured at 3, 6, and 24 h of duration. All formulations were run in triplicates, and average values were presented. The swelling ratio was calculated using the following Equation (3).
Swelling ratio (%) = (W2 − W1)/(W1) × 100(3)
where W1 is the weight before swelling and W2 is the weight after swelling

#### 2.5.6. Moisture Loss Study

The moisture retention capacity of the hydrogels was evaluated using the desiccant method of Standard Test Method for Water Vapour Transmission of Materials (ASTM E96/E96M-16, 2015) [26]. Hydrogel samples (0.5 g, triplicates) were spread in thin layers in a 2 cm-diameter crucible. The samples were then placed into airtight containers lined with a bed of silica gel as a desiccant. After 24 h, the crucibles containing the samples were re-weighed and percentage of moisture loss was calculated using the following Equation (4).
Moisture Loss (%) = (W2/W1) × 100 (4)
where W1 is the weight of the crucible before drying and W2 is the weight of the crucible after drying. Moisture loss was the water loss from the exposed surface of hydrogels over 24 h. The test was conducted on the CS, CS-NCC, CS-PL, CS-NCC-PL hydrogels. Intrasite™ gel served as the positive control.

#### 2.5.7. In Vitro Protein Release Assessment

The in vitro release profile of PL was investigated on two hydrogel preparations: CS-PL and CS-NCC-PL. Protein release was studied using Franz diffusion cell (Permegear Inc., Hellertown, PA, USA) in triplicates. All samples were prepared by mixing 5 mg/g of PL in hydrogel, and the proteins were incorporated into the hydrogel overnight at −10 °C. The release study was conducted using a cellulose acetate membrane with a mesh size of 0.45 μm. DI at pH 7.0 was used as the receptor medium. The receptor chamber was filled with DI until the mark of the sampling port was reached. The cellulose acetate membrane was then placed on top of the receptor chamber opening (0.7855 cm^2^). The receptor chamber was maintained at 37 ± 1 °C in a circulating water bath. The hydrogel weighing 1 g was placed on top of the cellulose acetate membrane, and all of the compartments were held together with a clamp. Samples (1 mL) were withdrawn at predetermined time intervals at 3, 6, and 24 h. The receptor medium was replaced with an equal volume of DI to maintain the sink conditions. The released proteins in the media were added with a few drops of Bradford reagent and assayed using a UV-Vis spectrophotometer (Shimadzu, Kyoto, Japan) at 280 nm. DI served as a blank. The absorbance of each sample was determined and recorded. The concentration of unknown proteins was then calculated by plotting the calibration curve. The cumulative amount of proteins that permeated out of the membrane was calculated and plotted against time (h). This experiment was repeated in triplicates for the in vitro release and characterization of therapeutic proteins present in PL that were incorporated in the hydrogel. A validated UV spectrophotometric method was used to quantify the PL loaded in the hydrogel systems at 280 nm to determine the concentration of proteins released at 3, 6, and 24 h.

### 2.6. Cytocompatibility Studies

#### 2.6.1. Isolation of Human Dermal Fibroblast (HDF)

This study was approved by the Universiti Kebangsaan Malaysia Research Ethics Committee (UKM PPI/111/8/JEP-2021-052). Redundant abdominoplasty skin tissue samples were received from the patient with written informed consent and processed within 48 h with the ISO protocol followed at CTERM. The skin was completely rinsed in DPBS, cut into small pieces (1–2 cm^2^), and immersed overnight in 10 mL of serum-free Epilife medium containing 25.3 mg of Dispase at 2–8 °C to isolate the epidermis from the dermis layers. On the next day, the epidermis layer was separated from the dermis layer. Thereafter, the dermis was chopped into smaller pieces and digested with 0.6% collagenase type I (Worthington, Columbus, OH, USA) for 4–8 h in an incubator shaker maintained at 37 °C. The cell suspension was centrifuged at 5000× *g* rpm for 5 min at 37 °C, and the cell pellet was rinsed with DPBS after trypsinization with TE-EDTA (0.05%). Finally, the cell pellet was re-suspended in F12: DMEM (1:1) supplemented with 10% FBS, 1% antibacterial–antimycotic, 1% Glutamax, and 2% HEPES. Cells were cultured at 37 °C in 5% CO_2_ with the medium changed every 2–3 days.

#### 2.6.2. Cell Culture

HDFs were used to study the cytocompatibility of the hydrogels. Cells with the passage of P1-3 were cultured in F12: DMEM supplemented with 10% FBS at 37 °C in an incubator supplemented with 5% CO_2_. PL (10%) was used alone, as well as in all hydrogel formulations. The culture medium was replenished every 2–3 days and subsequently trypsinized once more than 90% confluence was achieved and transferred to T75 flasks for further assay.

#### 2.6.3. Cell Viability

The cytotoxic effect of the fabricated hydrogels (CS, CS-NCC, and CS-NCC-PL) on skin cells was investigated. Cell viability assay was performed indirectly using primary HDFs (CTERM, UKM Medical Centre, Kuala Lumpur, Malaysia) cultured in DMEM by utilizing Alamar Blue. All cells were maintained at 37 °C in humidified 5% CO_2_ atmosphere. In this study, all hydrogel samples were washed with sterile DPBS and then incubated in a culture medium for 24 h to obtain the leachates (membrane sterilized 0.22 μm) from the hydrogel. HDFs with passage number P2 were seeded onto a 96-well plate at a density of 10^4^ cells/well and incubated in fresh media for 24 h at 37 °C to ensure cell attachment and proliferation. Leachates from the hydrogels were sterilized by membrane filtration (0.45 μm). Afterward, the culture medium was replaced with leachates of the hydrogels of volume 200 μL in each well. The microplate was then incubated for 24, 48, and 72 h, and 10% of Alamar Blue was added at the end of each time points and incubated for 4 h in the dark. Finally, OD was measured at 570 nm using a microplate reader (BioTek PowerWave XS, Winooski, VT, USA). The results were calculated as percentage cell viability relative to the control group (cells without hydrogel treatment) from Equation (5).
Cell viability (%) = [OD of treated/OD of control] × 100(5)

#### 2.6.4. Cell Proliferation

All hydrogel samples were placed in a micro well plate and washed with sterile DPBS prior to analysis. Thereafter, culture media were added and then incubated for 24 h to retrieve the extract from the hydrogels indirectly as previously mentioned in cell viability assay. HDFs (1 × 10^4^) were seeded in a 96-well plate for 24 h and subsequently treated with sterile hydrogel extracts. DMEM was used as control and incubated for 24, 48, and 72 h. Alamar Blue cell proliferation assay was carried out as per the manufacturer’s protocol in accordance with previous work [27]. Alamar Blue (10%) was added into each well containing the control and treatment groups and incubated at 37 °C for 4 h in the dark. Absorbance was recorded using a spectrophotometer (BioTek, PowerWave XS, USA) at 570 and 600 nm. The percent reduction in Alamar Blue was calculated from following [Equation (6)].
Percent reduction (%) = [(εOX) λ2Aλ1 − (εOX) λ1Aλ2]/[(εRED) λ1A’λ2 − (εRED) λ2A’λ1] × 100(6)
where (εOX) λ2 = 117,216, (εOX) λ1 = 80,586, (εRED) λ1 = 155,677, (εRED) λ2 = 14,652. Aλ1 and Aλ2 = Observed absorbance reading for the test well at 570 nm and 600 nm, respectively. A’λ1 and A’λ2 = Observed absorbance reading for control well at 570 and 600 nm, respectively.

#### 2.6.5. Scratch Wound Assay

The in vitro wound scratch assay is an economic and fast method to predict the wound healing ability of compounds by assessing their migration rate on the skin cells. In this work, the migration rate of the HDFs was calculated via scratch assay to ensure that the hydrogel formulations do not interfere in wound healing. The HDFs were seeded in a 12-well plate (Greiner Bio-One, Kremstest, Austria) and then incubated at 37 °C in humidified 5% CO_2_ atmosphere until 100% confluence. The spent culture was discarded, and a scratch was made at the middle of each well on a cell monolayer using a sterile 10 μL pipette tip. Afterward, the cells were rinsed with DPBS by slight swirling the microplate, and different hydrogel extracts were subsequently added to the scratched cells. Cells without treatment served as the control group. Wound closure was observed through live imaging by acquiring images every 60 min for 72 h at three spots per well using a Nikon A1R-A1 CLSM. Cell migration rate was calculated for 24 h. The tissue-culture plate inclosing cells was fixed inside the Chamlide Incubator System (Live Cell Instrument, Seoul, Korea) at 37 °C and 5% CO_2_. The images were analyzed by NIS Elements AR 3.1 (Nikon). The migration rate of the cells was calculated from following [Equation (7)].
Cell migration rate = (measurement at 0 h − measurement at 24 h)/24 h(7)

#### 2.6.6. LIVE/DEAD^®^ Cell Viability Assay

This assay was conducted to evaluate the functional status of the cells by identifying cytoplasmic esterase activity using the LIVE/DEAD™ Viability/Cytotoxicity kit for mammalian cells (Invitrogen). The kit comprises of calcein, which fluoresces green in living cells, and ethidium bromide, which fluoresces red in dead cells. In brief, the HDFs were plated at the same seeding density as per cell viability and proliferation assay and maintained as above prior to treatment with samples. The cells were treated with calcein and ethidium bromide for 30 min as per the manufacturer’s instructions. Later, the cells were rinsed with DBPS and observed using a Nikon A1R fluorescence microscope (Nikon, Tokyo, Japan).

#### 2.6.7. CS Stabilizing Effect on PL

A cell proliferation assay was carried out using protease degradation and heat treatment approaches to establish the protective effect of CS on PL.

##### Protease Degradation Test

Cell proliferation assay was conducted to determine the protective effect of CS against proteases on GF. In brief, all PL-loaded CS hydrogels and PL alone (10%) were subjected to 0.05% trypsin treatment at 37 °C for 1 h. Then, the HDFs (P2, 2 × 10^4^) were incubated with trypsin-treated PL and CS hydrogel for 24 h followed by Alamar Blue addition to the cells and incubated for the next 4 h. Then, absorbance was recorded at 570 and 600 nm. Cell proliferation was calculated by percentage reduction in Resazurin. Cells without treatment served as control.

##### Heat Treatment Test

PL (10%) and CS hydrogels containing PL were exposed to 0, 25 °C, 37 °C, and 45 °C for 24 h to examine the stabilizing effect of CS on PL. Then, extracts of both heat-treated groups were added to the HDFs (P3, 2 × 10^4^) as per above method, and cell proliferation was determined.

### 2.7. Statistical Analysis

Experiments were performed in triplicates and mean ± SD was reported. Data were analyzed using one-way analysis of variance and Tukey’s multiple comparisons test, by using GraphPad Prism version 5.00 (GraphPad Software, La Jolla, CA, USA). The level of significance was set at *p* < 0.05.

## 3. Results

### 3.1. Quantification of GFs in PL

The concentrations of TGF-β and EGF from PL were measured using ELISA. ELISA was performed for standard and sample as per manual instructions, and concentrations were determined as 477.26 pg/mL and 150 ng/mL for EGF and TGF-β, respectively, from the calibration curves. The level of TGF-β was threefold higher than that of EGF, which is consistent with reported literature [28].

### 3.2. Chemical Composition, Fiber Yield, Zeta Potential, and Crystallinity of NCC

After various stages of chemical treatment on raw, alkali-treated, and bleached fibers, chemical composition of the fibers was determined as previously described by Tuerxun Duolikun (2018). The results are presented in Table 1. After a series of chemical treatment on the kenaf biomass fibers, cellulose content significantly increased from raw to bleached to 30% to 84%, whereas hemicellulose and lignin contents declined to 5% and 3%, respectively [29].

The fiber yield of the NCC after acid hydrolysis was 40% (of initial weight), which is consistent with the fiber yield of NCC from other plant sources, such as sisal (30%) and mengkuang leaves (28%) [30]. In general, fiber yield is dependent on pre-treatment methods and hydrolysis environment. The low fiber yield of NCC might be caused by sulfuric acid treatment during production, which caused the degradation and removal of amorphous and other non-cellulosic regions of the fibers that result in weight loss.

The zeta potential of the NCC suspension was recorded as (−10.9 ± 5.47 mV). The NCC becomes crystalline and possesses a negative charge on the surface of the cellulose chain. They become more stable through electrostatic repulsion among the negatively charged groups on the polymer chains [31]. The anionic NCC is found suitable to form composite hydrogels with cationic CS polymer. The abundance of hydroxyl groups in NCC cellulose is responsible for the negative charges that bonded with CS via the electrostatic interaction of protonation of NH_2_ on CS and hydroxyl groups. Platelets also carry negative charges onto their surfaces due to the presence of sialic acid (N-acetyl-neuraminic acid) and amino acids such as glutamate and aspartate [32]. PL was loaded into NCC-reinforced CS hydrogel and associated by non-covalent bonding and subsequently stabilized by CS through protein bindings that promote fibrin gel formation and thus retained GF functionality for a long time [9].

Cellulose naturally comprises of crystalline and amorphous portions. The amorphous region (contains impurities of lignin, hemicellulose etc.) needs to be eliminated to make cellulose highly pure and crystalline with desirable properties. Therefore, chemical treatment including sulfuric acid hydrolysis was carried out to obtain an extremely crystalline and purified form of cellulose by hydrolyzing the amorphous region.

To investigate the crystallinity of the bleached fibers and effect of acid treatment on the resulting NCC, X-ray, diffractometry (XRD) was carried out. Figure 1 presents the obtained XRD patterns for kenaf bast fibers for bleached and acid treated fibers. The crystallinity index of both fibers were calculated using Diffrac.EvaV4.0 software. Diffraction peaks at 2θ value of 14.5°, 16.5°, and 22.5° at plane of 101, 10-1, and 002, respectively, were observed. From the diffraction pattern it could be noticed that cellulose was present in the form of cellulose [33,34]. Crystallinity index was calculated for the bleached fibers and NCC. For the kenaf biomass, the crystallinity index values of the bleached fibers and NCC were reported as 56% and 71.6%, respectively, in consistence with similar findings [35]. The higher crystallinity of NCC than the bleached fibers could be due to the removal of amorphous cellulosic region by acid hydrolysis. The XRD result suggested that the amorphous region of the bleached fibers degraded during extraction, whereas the crystalline region remained unaffected. The high crystallinity of NCC was related to the high tensile strength of the fibers. Therefore, the mechanical properties of the nanocomposite can be improved by using NCC as a reinforcing agent.

### 3.3. Scanning Electron Microscopy

Surface morphological analysis was performed on the bleached fibers and kenaf NCC. The SEM images are depicted in the Figure 2 at different magnifications. As shown in Figure 2a,c the bleached fiber bark structures were apparent. Meanwhile, NCC microphotographs in Figure 2b,d revealed that the surface of the NCC became smooth and the fibers appeared as a web-like network with diameters approximately 40–90 nm in consistent with similar SEM surface morphologies [24]. After a series of chemical treatments, impurities such as pectin, hemicellulose, and lignin were removed from the structure of the bleached fibers, as can be implied from the FTIR findings. The small diameter of NCC was because of acid hydrolysis, which significantly removed the amorphous portion from the crystalline part, which is in accordance with previous similar finding [34].

### 3.4. FTIR-ATR Spectroscopic Analysis of NCC

FTIR was carried out on raw fibers, bleached fibers, and NCC to determine their chemical composition after chemical and mechanical treatment. The results are shown in Figure 3a and Table 2. All spectra detected a broad and intense peak at 3300 cm^−1^ region which is attributed to the characteristic of polysaccharides hydroxyl bonds [36]. C–H symmetrical stretching and CH2 symmetrical stretching at 2900–2800 cm^−1^ revealed polysaccharide, wax, and oil content of the fibers [36]. Another peak was observed in the spectra of the raw fibers at 1242 cm^−1^, which was associated with C-O stretching of the aryl group present in lignin. The disappearance of this peak in bleached and NCC suggested the separation of lignin by chemical treatment [36].The vibration peak in the fibers at 1326–1340 cm^−1^ was linked to bending of the C-H and C-O bonds in polysaccharide aromatic rings [36]. Absorption peak was shown in all samples at 1023–1030 cm^−1^, which were associated with C-O and C-N vibrations [36]. The absorption peak at 1634 cm^−1^ in NCC was due to adsorbed water, which was suggestive of NCC [36].

FTIR was conducted to investigate the possible interaction of functional group between PL and CS hydrogel, as presented in Figure 3b. CS showed a broad band at 3500–3200 cm^−1^, which can be attributed to the O–H and N–H stretching vibrations of functional groups in hydrogen bonds. Characteristic absorption bands at 1634 cm^−1^ corresponded to C=O stretching in the amide I vibration. The absorption band at 1540 cm^−1^ appeared due to N–H bending in amide II vibration. The spectra at 1069 and 1024 cm^−1^ corresponded to C–O stretching vibrations, which were characteristics of CS structure. Association of PL in CS hydrogel was confirmed by displacement of characteristic bands at 3219–3270 cm^−1^ due to O-H stretching and 1634–1641 cm^−1^ due to NH vibrations [37]. These modifications suggested a possible interaction between PL and CS.

FTIR spectra of NCC, CS, PL, and CS-NCC-PL are represented in Figure 3c. A characteristic peak of CS was observed at 3310 cm^−1^ owing to the O-H group. CS-NCC was observed by shifting of the peak from 3337 cm^−1^ to 3759 cm^−1^, which indicated possible overlapping of hydrogen bonds. PL with a characteristic peak at 3885 cm^−1^ was due to O-H stretching that shifted slightly toward 3901 cm^−1^. Thus, CS incorporated with NCC and PL possessed two extra peaks, which corresponded to NCC and PL as compared with control CS hydrogel. These results indicate that NCC was prepared and successfully incorporated in CS hydrogel along with PL.

### 3.5. Swelling Study

Freeze drying allows the nucleation of ice crystals from solution and further growth along the lines of thermal gradients. Exclusion of the CS acetate salt from the ice crystal phase and subsequent ice removal by lyophilization generate a porous material [38]. Freeze drying is usually employed to produce porous CS hydrogels to assess their water swelling behavior effectively.

An ideal dressing must be capable of absorbing the wound exudates that could potentially cause bacterial infection at the wound bed. Swelling properties determines the moisture absorption capacity of a dressing, which is a crucial action for absorbing pus and exudates from weeping wounds. The swelling ratios over time of the freeze-dried hydrogels are shown in Figure 4. All hydrogels swelled at different rates, and equilibriums were achieved up to 1000 times of their initial weight within 24 h of water imbibition. During the initial swelling, water was absorbed via capillaries located in the internal structure of the hydrogels. As 3D networks, porous structures provide channels for the molecules to enter and escape. The hydrophilic groups (-OH/-COOH/-COO-) present in CS were bound with water molecules by hydrogen bonds that created a hydration layer. This phenomenon may explain why all hydrogels showed the fastest swelling at the beginning (<3 h). The addition of PL in the CS hydrogels reduced the swelling ratio (~500% reduction). The addition of PL may decrease availability of CS hydrophilic groups to bind with water. The swelling ratio also decreased further (~300% reduction) when NCC was added into CS. This behavior could be ascribed to the fact that NCC lodges the free space volume in the CS polymeric matrix, thereby restricting the volume available for swelling and causing the formation of a rigid hydrogel structure that cannot be easily penetrated by water molecules. Hence, the water absorption decreased, which consequently decreased the swelling ratio [39]. However, the addition of PL to the CS-NCC hydrogel increased the swelling ratio owing to its large hydrophilic groups in molecular chains, which promoted the establishment of hydration layers [40]. The higher water holding capacities (>1000%) of the hydrogels indicate that the formulated hydrogels are suitable for medium to heavy suppurating wounds. This is a crucial property for a dressing in minimizing infection. At the same time, the hydrogels maintain a moist local microenvironment for tissue repair process to take place.

### 3.6. Moisture Loss Study

Moisture retention capacity is an important characteristic for a wound dressing. Hydrogels should provide a moist environment to the wound to accelerate healing [40]. Moisture loss from the different hydrogels was evaluated by employing the desiccation method. Moisture loss was expressed in percentage and calculated after 24 h of hydrogel drying. As shown in Figure 5, Intrasite™ hydrogel lost almost 80% of moisture after 24 h, followed by PL, which lost 90% of moisture. Meanwhile, the CS-PL hydrogel lost approximately 42% of moisture as compared with the CS hydrogel. The PL entrapped within the network of polymer and prevented moisture loss. Therefore, the hydrogels under investigation held moisture for a longer period as compared with the commercial hydrogel. The CS-NCC-PL hydrogel can maintain a moist environment on chronic wounds for an extended time.

### 3.7. In Vitro Protein Release Assay

The cumulative release profiles of PL from the CS-PL and CS-NCC-PL hydrogels are represented in Figure 6. The PL released from the CS-PL hydrogel showed faster release at the first 3 h compared with that from CS-NCC. Thereafter, the PL release increased gradually (PL flux = 0.165 mg/cm^2^/h) and reached the maximum release of 5.22 ± 1.47 mg/cm^2^ at 24 h. During this time, proteins permeated from the inside matrix of the hydrogel via swelling and diffusion mechanism. In the CS-NCC-PL hydrogel, the rate of PL release was significantly decreased and much controlled throughout the test duration (PL flux = 0.075 mg/cm^2^/h). This phenomenon suggested that NCC might act as a nanofiller and provide sustained release because of its capacity to retain the proteins within the hydrogel matrix for a long period of time (maximum PL release of 3.00 ± 1.11 mg/cm^2^ at 24 h). For chronic wound healing, hydrogel comprises of platelet proteins, and a cocktail of various GFs is desired for tissue proliferation. The CS-NCC hydrogel composite is successful in controlling the PL release and is a promising vehicle for PL in wound healing.

#### 3.7.1. Cell Viability

The cytotoxic effect of hydrogel formulations on HDFs was tested by Alamar Blue assay. The results of cell viability are presented in Figure 7. This experiment revealed that the cell viability of the CS-PL and CS-NCC-PL hydrogels was higher than that of PL alone for 24, 48, and 72 h. This hike in cell viability of the PL-loaded hydrogels is probably due to the protective effect of CS-NCC on PL. As mentioned previously, CS can stabilize PL and protect it from degradation. Overall, the cytotoxic effect of hydrogels in this study with or without PL was negligible (*p* < 0.05, one-way ANOVA, Tukey’s multiple comparison test). Therefore, the hydrogel formulations are non-toxic and safe to be applied onto wounds.

#### 3.7.2. Cell Proliferation

Cell proliferation was analyzed through Alamar Blue assay to ascertain any change in cell proportion. In this study, CS-NCC-PL hydrogels were tested by treating HDFs (1 × 10^4^) with sterile hydrogel leachates in the microplates for 24, 48, and 72 h. The results of cell proliferation are provided in Figure 8. PL alone showed low proliferation with an increasing trend over 3 days. The hydrogels enhanced the HDF proliferation from 24 h to 48 h, demonstrating the growth and proliferation of the HDFs. The PL-loaded hydrogels significantly augmented fibroblast proliferation (>100% viability) compared with the blank hydrogels or PL alone, indicating fast wound closure without causing any cell toxicity. Higher cell proliferation induced by combination of chitosan with PL is also in agreement with similar study where it showed that platelet-rich plasma with chitosan-induced growth factor enrichment can stimulate the growth of fibroblasts [41]. This dictates the feasibility and safety of autologous PL in wound healing and complications in patients with intractable diseases such as diabetic ulcers and decubitus [41].

### 3.8. Wound Scratch Assay

Cell migration across the provisional gap was carried out to assess the healing potential of the hydrogels. The results in Figure 9 show the effect of hydrogels on the migration rate of the HDFs for 24 h. All hydrogel groups showed significant migration of cells as compared with the control group (*p* < 0.05, one-way ANOVA followed by Tukey’s multiple comparison test). The migration rates of the HDFs after being treated with the CS-PL hydrogels was significantly higher than CS alone (*p* < 0.05). This indicates the PL stimulates the cell migration rate. The addition of NCC does not affect the migration rate as CS-NCC-PL and CS-PL group (*p* > 0.05). The wound area-time plot and micrographs (Figure 10 and Figure 11) indicated that all wounds were nearly closed in less than 48 h despite of the differences in the migration rates of the control and treatment groups. The HDFs were managed to occupy the free space within a span of 48 h, indicating that the hydrogels efficiently healed the HDFs.

### 3.9. LIVE/DEAD^®^ Cell Viability Assay

Cytotoxic effect of hydrogel formulations was further investigated using LIVE/DEAD™ Cell Viability Assay. This assay gives qualitative aspect of cell viability. The control and cells treated with the hydrogels lacked dead cells (stained red), as shown in Figure 12. Control and chitosan hydrogels with different combinations exhibited favorable cell viability after 24 h of incubation (captured in green color cells). In addition, after 48 h, cells continue to grow and become elongated for further maturation and differentiation. Additionally, these results comply with the results of Alamar Blue™ assay, suggesting the non-toxicity of the hydrogel after 24 and 72 h of treatment. Cell number and cell viability increased over the period of 72 h.

### 3.10. CS Stabilizing Effect on PL

Cell proliferation assay was carried out to assess the protective effect of CS on GF present in PL. CS along with NCC hydrogel with and without PL and PL alone groups were subjected to heat treatment and protease digestion. Afterward, the HDFs were subjected to these treatments for 24, 48, and 72 h. Results of these findings are provided in Figure 13a,b. All data show the significant protective effect of CS on PL for proteases and heat treatment. The PL loaded with CS or CS-NCC over 72 h had significantly higher proliferation than those unprotected PL. This protective effect of chitosan is supported with similar work carried out by Fisher et al., that proposed the mechanism where chitosan fibers tightly bind with major plasma proteins and a specific sub-set of platelet surface proteins which results in the acceleration of fibrin gel formation when platelet integrins contact with plasma proteins. This result indicates that CS can protect the GF present in PL, thereby promoting fast wound healing.

## 4. Conclusions

In general, GFs in PL suffer clinical limitations because of rapid degradation by proteases at the tissue site. To address the challenges related to PL instability, we developed a CS hydrogel combined with kenaf-derived NCC. It can stabilize PL and control the release of PL onto the wound site for prolonged action. Kenaf biomass-derived NCC at 0.4% concentration acted as the nanofiller for CS hydrogel and controlled PL release at a slower rate (maximum PL release of 3.00 ± 1.11 mg/cm^2^ at 24 h) compared with the CS hydrogel alone. FTIR study confirmed the presence of NCCs and PL in the CS matrix. Swelling data revealed that NCC incorporation in the PL-loaded CS hydrogel possesses high swelling ratio and water retention capacity (>1000 times at less than 3 h), thereby benefiting the healing of high exudate wounds. In vitro study revealed that the hydrogels are non-toxic to host tissues (>100% HDF cell viability) and are able to close wounds at a faster rate compared with the controls. The protective effect of CS upon GFs in PL was also demonstrated via protease and heat treatment of the hydrogels. Thus, NCC-reinforced CS hydrogel emerged as a promising PL vehicle especially for autologous wound therapy for chronic wounds. This new hydrogel PL carrier may be extended to other autologous PRP therapies, such as rheumatoid arthritis, bone regeneration, low-grade musculoskeletal injuries, and dental applications.

## Figures and Tables

**Figure 1 polymers-13-04392-f001:**
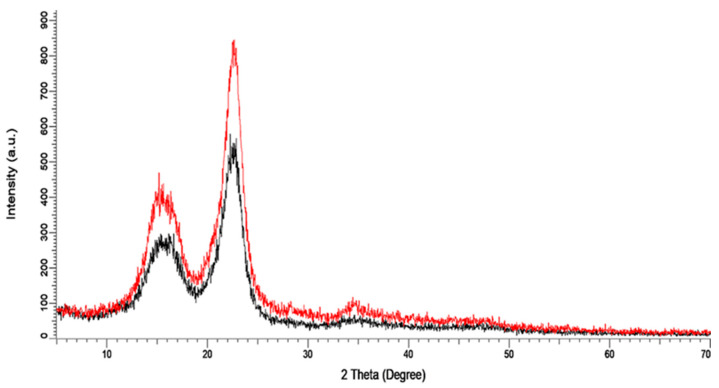
The XRD patterns of both NCC (Red) and (Bleached fibers (Black) from kenaf bast fiber.

**Figure 2 polymers-13-04392-f002:**
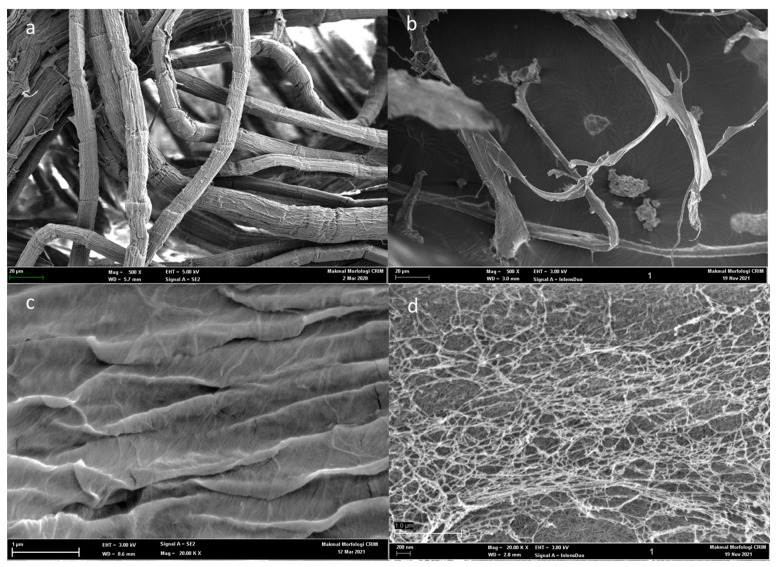
Photographs of SEM images of (**a**) bleached fibers (**b**) NCC at 500X (**c**) bleached fibers (**d**) NCC at 20KX.

**Figure 3 polymers-13-04392-f003:**
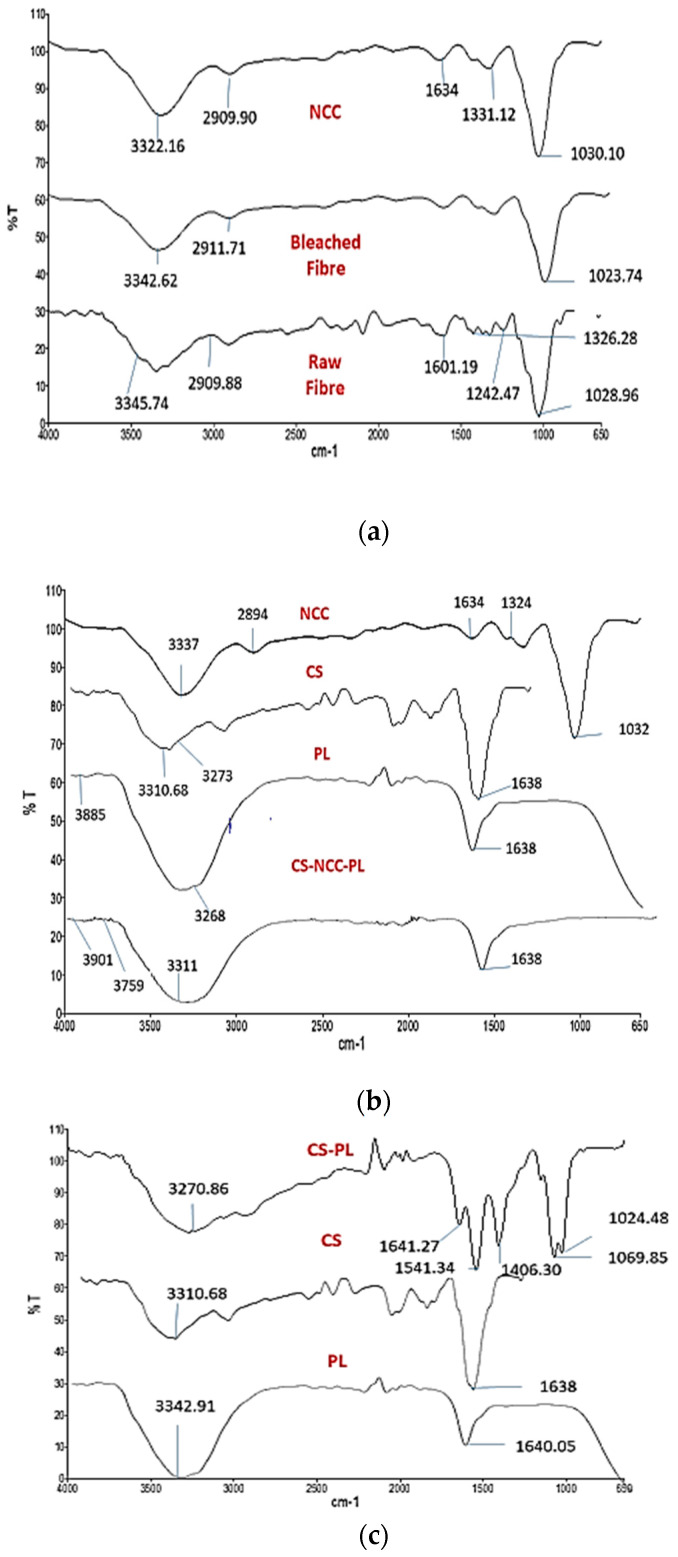
FTIR analysis comparison of (**a**) NCC, bleached fiber and raw fiber (**b**) NCC, CS, PL, and CS-NCC-PL (**c**) CS-PL, CS and PL alone.

**Figure 4 polymers-13-04392-f004:**
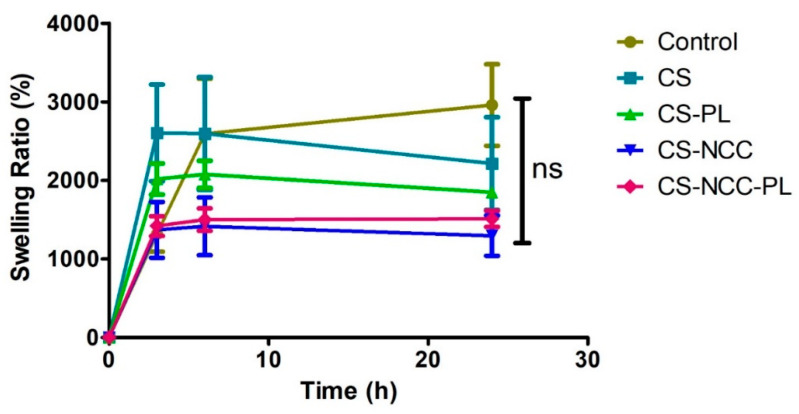
Swelling ratio of Control (Intrasite Gel), CS hydrogel, CS-NCC hydrogel, CS-PL hydrogel and CS-NCC-PL hydrogel over 24 h. The error bars showed standard deviation based on triplicated samples at each time point.

**Figure 5 polymers-13-04392-f005:**
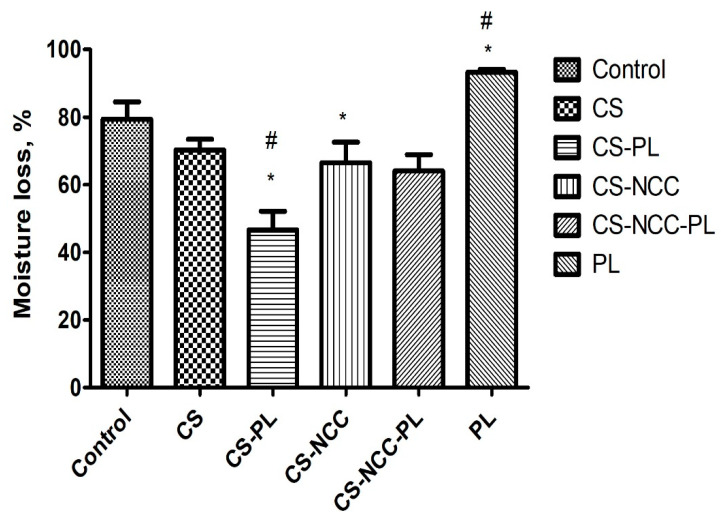
Moisture loss profile of formulated hydrogels over 24 h (expressed in mean ±SD, *n* = 3). The asterisks (*) represent significant difference (*p* < 0.05) compared to control. The hashtag (#) represent significant difference (*p* < 0.05) between the groups.

**Figure 6 polymers-13-04392-f006:**
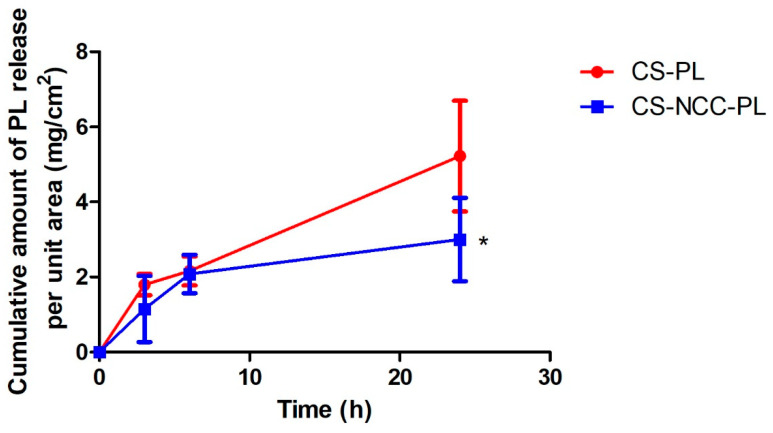
In vitro cumulative release profile of PL from CS-PL and CS-NCC-PL. (*) means significant difference (*p* < 0.05) compared to CS-PL.

**Figure 7 polymers-13-04392-f007:**
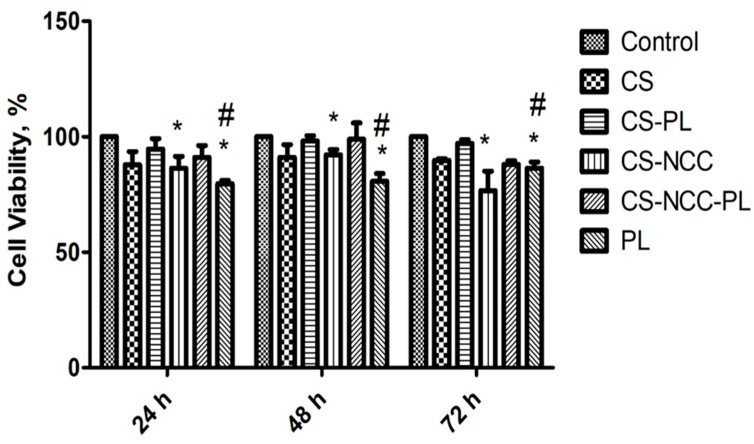
Percentage cell viability of hydrogels on Human dermal fibroblast (HDF) at different time points using Alamar blue assay, *n* = 3, The asterisks (*) represents statistically significant difference compared to control (*p* < 0.05) and hashtag (#) represent significant difference (*p* < 0.05) between groups.

**Figure 8 polymers-13-04392-f008:**
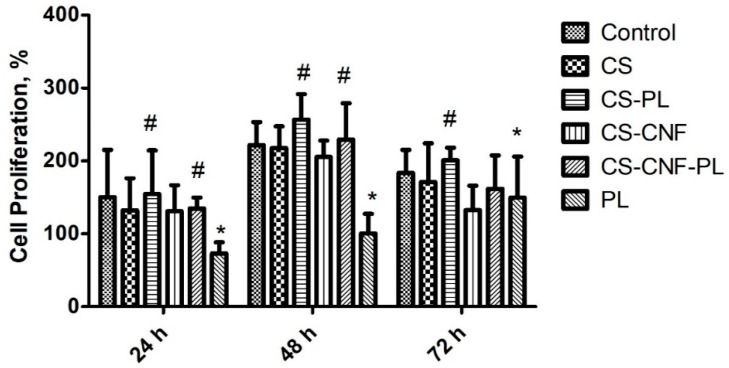
Cell proliferation study of hydrogels on human dermal fibroblast (HDF) at different time points using Alamar blue assay, *n* = 3, The asterisks (*) represent statistically significant different (*p* < 0.05) compared to control, the hashtag (#) means statistically significant different (*p* < 0.05) between groups.

**Figure 9 polymers-13-04392-f009:**
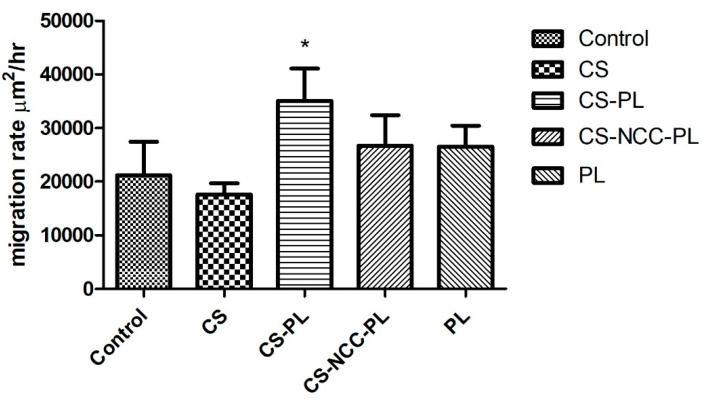
Rate of cell migration of HDFs treated with different hydrogels; *n* = 3, *—statistically significant different (*p* < 0.05) compared to CS and control.

**Figure 10 polymers-13-04392-f010:**
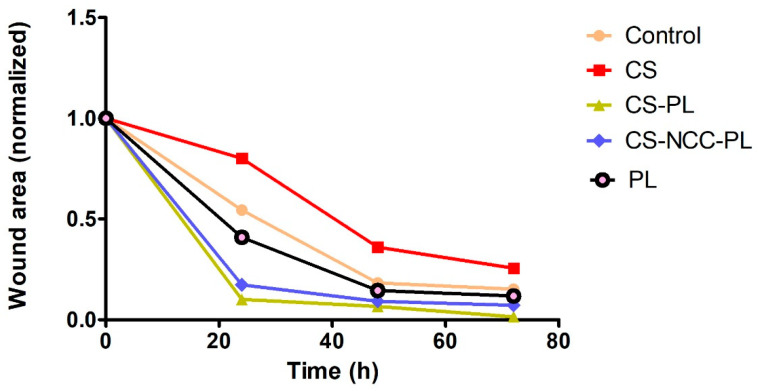
Wound area closure at different time points; *n* = 3.

**Figure 11 polymers-13-04392-f011:**
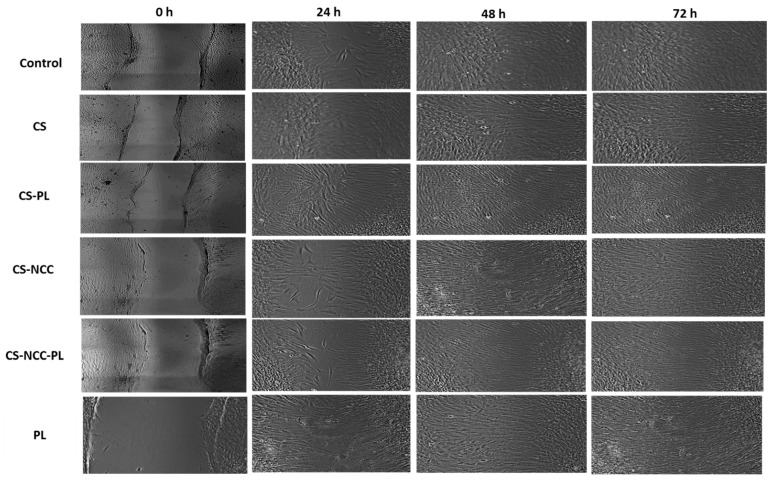
Migration of HDFs treated with hydrogels for 24, 48, and 72 h.

**Figure 12 polymers-13-04392-f012:**
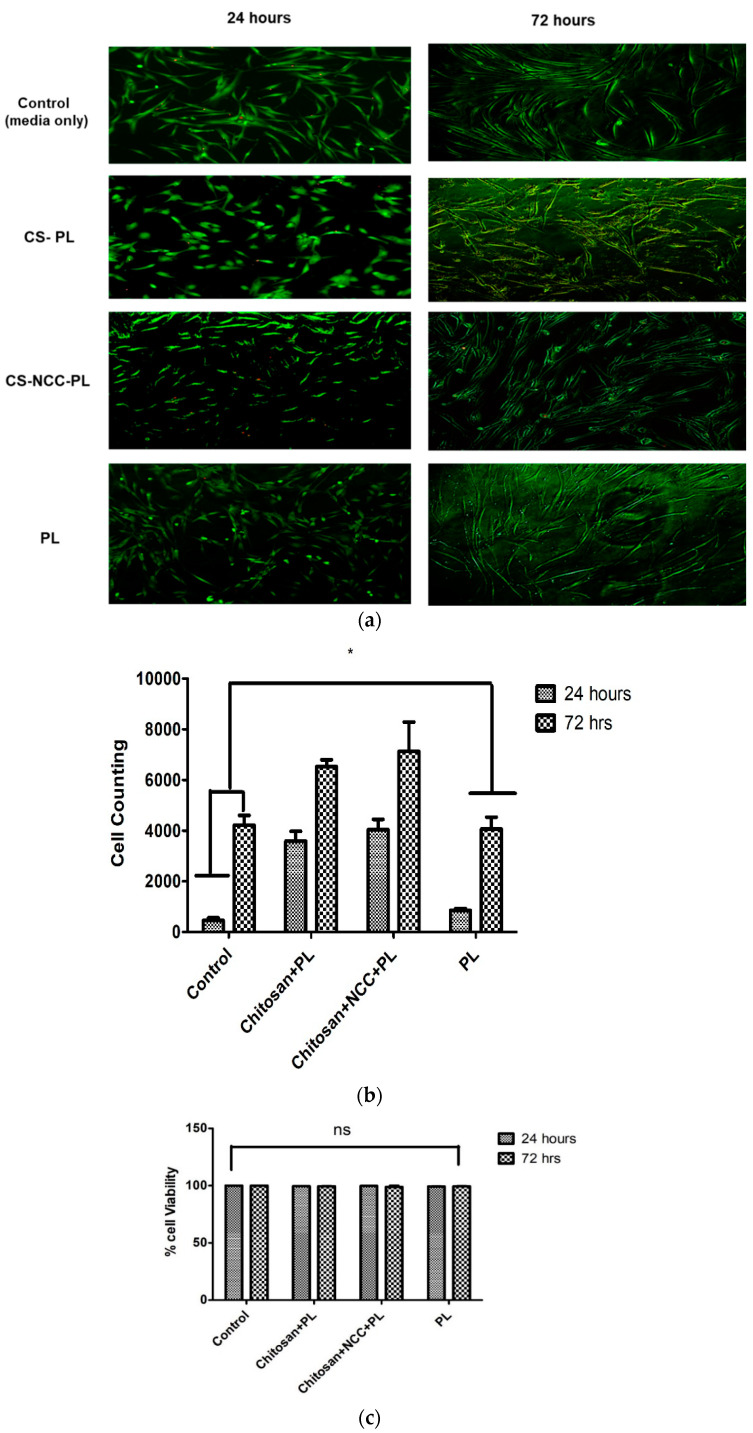
(**a**) Live Dead cell viability Assay at 200X magnification, (**b**) Cell counting treated with different hydrogels at 24 h and 72 h (**c**) % cell viability compared to control. *n* = 3. The asterisks (*)—statistically significant different (*p* < 0.05); ns—non-significant from control (untreated cells).

**Figure 13 polymers-13-04392-f013:**
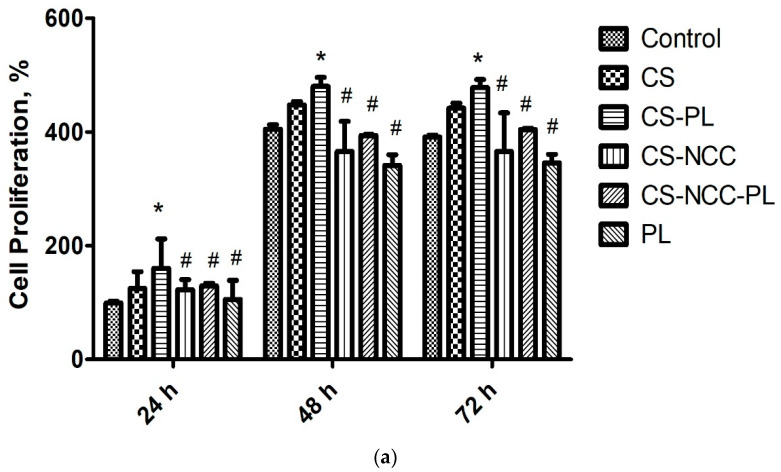
Chitosan stability studies on GF present in PL by cell proliferation study using Alamar blue assay (**a**) Effect of proteases treatment (**b**) Effect of heat treatment, *n* = 3, The asterisks (*) —statistically significant different compared to PL alone (*p* < 0.05); The hashtag (#) means statistically significant different (*p* < 0.05) between groups.

**Table 1 polymers-13-04392-t001:** Chemical composition of kenaf fibers after chemical treatments.

Material	Chemical Composition (%)		
	Cellulose	Hemicellulose	Lignin
Raw	30	31	30
Alkali-Treated	70	20	15
Bleached	84	5	3

**Table 2 polymers-13-04392-t002:** FTIR characteristic peaks of Kenaf raw fibers, bleached fibers, and NCC.

Raw Fibers Peak (cm^−1^)	Bleached Fibers Peak (cm^−1^)	NCC Peak (cm^−1^)	Peak Assignment
3345.74	3342.62	3322.16	O–H stretching
2909.88	2911.71	2909.90	C–H symmetrical stretching and CH2 symmetrical stretching
1326.28	-	-	C–O aromatic ring
1242.47	-	-	C–O stretching
1601.19	-	-	C=C groups
-	-	1634	Adsorbed water
1028.96	1023.74	1030.10	C-O and O-H groups stretching

## Data Availability

The data presented in this study are available on request from the corresponding author.

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
