# Peer review of "Chitosan Reinforced with Kenaf Nanocrystalline Cellulose as an Effective Carrier for the Delivery of Platelet Lysate in the Acceleration of Wound Healing"

_polymers, 2021, doi:10.3390/polym13244392_

Round 1

Reviewer 1 Report

The manuscript polymers-1453436 studies the preparation of a hydrogel composite based on chitosan and kenaf nanocrystalline cellulose, with the aim to stabilize platelet lysate and to control its release onto the wound site for prolonged time.

The manuscript is clearly presented, well written and I recommend the publication in Polymers journal, after major revisions:

  1. Materials and Methods

- L. 157: Please mention the name of the freeze-drying equipment!

- L. 198: Please add the corresponding unit of measurement (%) to Eq. (2)!

- L. 201: Please add the SEM abbreviation of scanning electron microscopy at the title of 2.5.3 section!

- L. 213: Please use „FTIR-ATR spectrometer” not „FTIR-ATR”!

- L. 224: Please add the corresponding unit of measurement (%) to Eq. (3)!

3.2. Chemical composition, fiber yield, zeta potential, and crystallinity of NCC

- L. 403-407: Actually, in a typical XRD diffractogram of cellulose I, can be identify five crystalline peaks, namely (101), (10-1), (021), (002) and (040), which appear at different Bragg angles, such as (approximately): 14º, 16º, 20º, 22º and 35º, respectively (Figure 1). However, the most known peaks are those of the (101), (10-1) and (002) crystalline planes. Moreover, the peak that is due to the amorphous cellulose is distributed in the range of 10-28º and has the maximum intensity at a Bragg angle of 18 – 19°, can be also 20°, but in no case at 16º, as was mentioned by the authors. The amorphous peak is a broad peak situated at the basis of XRD diffractogram. One of the methods used to determine the crystallinity of the samples is the subtracting of the amorphous contribution from diffraction spectra using an amorphous standard (this can be done also with the software of the XRD diffractometer). For the XRD peak height method, the authors must identify the amorphous peak and measure the maximum height of the amorphous peak, which should be situated between the (101) and (002) peaks. Please make the corresponding corrections related to the identification of the crystalline peaks characteristic to cellulose I and recalculate the degree of crystallinity, taking into account the correct position of the amorphous peak in the XRD diffractogram.

- It is a repetition of the same conclusion in both paragraphs, L.409-411 and L. 415-416. Please remove one of these sentences!

3.3. Scanning electron microscopy

- L. 434: SEM images (Figure 2) are not clear! Moreover, the magnification of the samples are not visible at all from the figures. Please improve the figures and the make visible the notations under the figures!

- L. 430: “were removed from the structure of the bleached fibers” not “were removed from the surface structure of the bleached fibers”!! Pectin, hemicellulose and lignin are removed from the whole fiber, not only from the surface of the fiber!

3.4. FTIR-ATR spectroscopic analysis of NCC

- L. 440: “is characteristic for stretching vibration of OH groups in cellulose” not “corresponded to free OH groups on cellulose molecules”!! This broad peak includes also bands characteristic to the intra- and intermolecular hydrogen bonding in cellulose!

- L. 477-478: There is a reversal between the figures 3b and 3c in the caption of Figure 3. Please make the correction!

- Please rearrange all figures from Figure 3, so that each characteristic band from FTIR-ATR spectra, for all samples, to appear at the same wavenumber (cm-1)! For example, in Figure 3b the bands at 1638 cm-1, has different positions for each sample (CS, PL, NCC, CS-NCC-PL), in the range of 2250 – 1500 cm-1! Moreover, there is a big difference between CS spectrum from Figure 3b and the same CS spectrum from Figure 3c. The authors cannot identify the differences between the FTIR spectra of the samples, if the spectra differ so much from one figure to another!

- L. 441: Please explain the meaning of each band: “the absorption band at 1601 cm-1 observed in raw fibers is related to the stretching of C=C bonds of aromatic rings, which can be associated with lignin” not “the absorption peak at 1601 cm−1 in the raw fibers was absent in the bleached fibers”!

- This paragraph should be reorganized in order to avoid the repetition! For example, the band at 3300 cm-1 from NCC spectra is mentioned at L. 439 and L. 462; the band at 1630 cm-1 is mentioned at L. 449 and L. 463-464. Please try to explain the differences between the spectra without mentioning the same sample two or three times!

3.5. Swelling study

- L. 485-486: The sentence is ambiguous! Please revise it!

- L. 508: The y-axis from Figure 4 needs a unit of measurement! For swelling ratio, the unit is %!

- L. 504-507: The paragraph is not clear! Please explain better the comparation between the hydrogels!

4. Conclusion

- L. 630: “Swelling data” not “Swelling readings”!

Author Response

Responses to Reviewers                                               

Manuscript Title: Chitosan Reinforced with Kenaf Nanocrystalline Cellulose as an Effective Carrier for the Delivery of Platelet Lysate in the Acceleration of Wound Healing”

Manuscript ID: polymers-1453436

Dear Reviewers, thank you for your constructive comments and input. Here are our point-to-point response to your queries (in bold). Changes in the manuscript are highlighted in red.

Reviewer #1

The manuscript polymers-1453436 studies the preparation of a hydrogel composite based on chitosan and kenaf nanocrystalline cellulose, with the aim to stabilize platelet lysate and to control its release onto the wound site for prolonged time.

The manuscript is clearly presented, well written and I recommend the publication in Polymers journal, after major revisions:

  1. 157: Please mention the name of the freeze-drying equipment!

Labconco, Model: 117, Weight: 24 Kg

  1. 198: Please add the corresponding unit of measurement (%) to Eq. (2)

It is now corrected (Line 198)

  1. 201: Please add the SEM abbreviation of scanning electron microscopy at the title of 2.5.3 section!

It is now corrected (Line 205)

  1. 213: Please use „FTIR-ATR spectrometer” not „FTIR-ATR”!

It is now modified  (Line 217)

  1. 224: Please add the corresponding unit of measurement (%) to Eq. (3)!

It is now modified (Line 227)

3.2. Chemical composition, fiber yield, zeta potential, and crystallinity of NCC

  1. 406: Actually, in a typical XRD diffractogram of cellulose I, can be identify five crystalline peaks, namely (101), (10-1), (021), (002) and (040), which appear at different Bragg angles, such as (approximately): 14º, 16º, 20º, 22º and 35º, respectively (Figure 1). However, the most known peaks are those of the (101), (10-1) and (002) crystalline planes. Moreover, the peak that is due to the amorphous cellulose is distributed in the range of 10-28º and has the maximum intensity at a Bragg angle of 18 – 19°, can be also 20°, but in no case at 16º, as was mentioned by the authors. The amorphous peak is a broad peak situated at the basis of XRD diffractogram. One of the methods used to determine the crystallinity of the samples is the subtracting of the amorphous contribution from diffraction spectra using an amorphous standard (this can be done also with the software of the XRD diffractometer). For the XRD peak height method, the authors must identify the amorphous peak and measure the maximum height of the amorphous peak, which should be situated between the (101) and (002) peaks.

Please make the corresponding corrections related to the identification of the crystalline peaks characteristic to cellulose I and recalculate the degree of crystallinity, taking into account the correct position of the amorphous peak in the XRD diffractogram.

Diffraction peaks of cellulose I is identified with their correct positions and crystallinity index is recalculated (Lines 406) with graph.

It is a repetition of the same conclusion in both paragraphs, L.409-411 and L. 415-416. Please remove one of these sentences!

The repetition is deleted

  1. 434: SEM images (Figure 2) are not clear! Moreover, the magnification of the samples are not visible at all from the figures. Please improve the figures and the make visible the notations under the figures!- not available

SEM images are improved (Figure 2)

  1. 430: “were removed from the structure of the bleached fibers” not “were removed from the surface structure of the bleached fibers”!! Pectin, hemicellulose and lignin are removed from the whole fiber, not only from the surface of the fiber!

Line 446 is modified.

3.4. FTIR-ATR spectroscopic analysis of NCC

  1. 440: “is characteristic for stretching vibration of OH groups in cellulose” not “corresponded to free OH groups on cellulose molecules”!! This broad peak includes also bands characteristic to the intra- and intermolecular hydrogen bonding in cellulose!

All spectra detected a broad and intense peak at 3,300 cm−1 region which is attributed to the characteristic of polysaccharides hydroxyl bonds. It is now corrected (L-444)

  1. 477-478: There is a reversal between the figures 3b and 3c in the caption of Figure 3. Please make the correction!

It is now corrected (Fig 3)

Please rearrange all figures from Figure 3, so that each characteristic band from FTIR-ATR spectra, for all samples, to appear at the same wavenumber (cm-1)! For example, in Figure 3b the bands at 1638 cm-1, has different positions for each sample (CS, PL, NCC, CS-NCC-PL), in the range of 2250 – 1500 cm-1! Moreover, there is a big difference between CS spectrum from Figure 3b and the same CS spectrum from Figure 3c. The authors cannot identify the differences between the FTIR spectra of the samples, if the spectra differ so much from one figure to another!

Figure 3: It is now corrected

  1. 441: Please explain the meaning of each band: “the absorption band at 1601 cm-1 observed in raw fibers is related to the stretching of C=C bonds of aromatic rings, which can be associated with lignin” not “the absorption peak at 1601 cm−1 in the raw fibers was absent in the bleached fibers”!

Table 2: It is now included to define the meaning of each band

This paragraph should be reorganized in order to avoid the repetition! For example, the band at 3300 cm-1 from NCC spectra is mentioned at L. 439 and L. 462; the band at 1630 cm-1 is mentioned at L. 449 and L. 463-464. Please try to explain the differences between the spectra without mentioning the same sample two or three times!

Paragraph is now reorganized (Lines 448)

3.5. Swelling study

  1. 485-486: The sentence is ambiguous. Please revise it.

It is modified (Line 510)

  1. 508: The y-axis from Figure 4 needs a unit of measurement. For swelling ratio, the unit is %

It is corrected (Figure 4)

  1. 504-507: The paragraph is not clear! Please explain better the comparation between the hydrogels!

It is now revised (L. 516)

  1. 530: “Swelling data” not “Swelling readings”!

It is corrected.

#Reviewer 2:

“to formulate a vehicle” this formulation is out of context. The abstract is to verbally and requires some brief quantitative results

It is now revised. (see abstract)

The same apply for conclusion

It is now revised. (see conclusion)

“Therefore, the use of kenaf-based NCC as wound dressing has yet to be explored.” This seems rather a commercial challenge than a scientific one

Agree. This research is also exploring the potential commercial use of kenaf in medical application such as wound dressing as kenaf is a widely cultivated crop in Malaysia.  

“R & M chemicals, UK “ company was difficult to identify  (please provide a link )

A space between value and units should be, i.e. “4°C” and many other please check the entire paper for this problem.

It is now checked and corrected (Whole document)

“-80°C” this is not very cold  temperature ?

This temperature is required to support the fact that preservation temperature for platelet lysate to maintain the functionality. Reference: (https://onlinelibrary.wiley.com/doi/full/10.1002/jor.24520)

Equations 2, 5 should be written with professional equation not a photo !

Equation 2 (Line-199), Equation 5 (Line-304)- It is now corrected

Define all the acronyms before their first appearance in text, i.e. “SEM” and so on

2.5.3 Scanning electron microscopy (SEM)-(Line-205)- It is now corrected and at other places as well.

Please provide the approval “This study was approved by the Universiti Kebangsaan

The approval number is inserted (L-267) and the approval letter is appended at the end of the document.

Provide details of “ISO protocol”

ISO Ref 2, Role of plasma-derived fibrin on keratinocyte and fibroblast wound healing.

What does means this “CTERM”?

It is stands for “Centre For Tissue Engineering & Regenerative Medicine”

How was controlled the humidity “in humidified 5%”?

The humidity of the CO2 incubator is not monitored. We make sure that the water level in the tray is sufficiently weekly to maintain a moist environment suitable for cell culture.

The statistical analysis is very poor presented, such as difficult to understand the evaluation process

It is now revised (Line-364)

“of PL must be optimum to exert” what does means optimum here?

The sentence is deleted.

The Image in Figure 2 are too fuzzy and difficult to evaluate them properly- better quality is required and also the scale bar should be really visible- I have only these SEM images for nanofibers

Figure 2 is improved with better resolution and visible scale bars.

“the swelling ratio owing to its large hydrophilic groups”- endorse it by a ref!

L529. Reference [40]: Xu Z, Tang E, Zhao H. An Environmentally Sensitive Silk Fibroin/Chitosan Hydrogel and Its Drug Release Behaviors. Polymers (Basel). 2019 Dec 1;11(12):1980. doi: 10.3390/polym11121980. PMID: 31805749; PMCID: PMC6960489.

“bound protein at or near the matrix surface that leached through the membrane”- endorse it by a ref! otherwise not sure from where come this assumption.

The sentence is deleted.

Figure 9 contain too many Images therefore I suggest splitting it in 3 Figure separately

Figure 9 is now split to Fig 9, 10 and 11.

Figure 9c first raw apart of last image( very down one) the image are the same ! and claiming different composition ! Please check this and all other images from this Figure !

Figure 9c is now Fig 11. The day zero images for control and PL are now revised.

Figure 9 a is very fuzzy and requires scale bar ! the same to many image for a Figure !

Figure 9 a is now Fig 9 is now improved with higher resolution. Scale bar could not be overlaid onto the images as scale bar option is not supported by the software (Nikon, NIS Elements AR 3.1)

Most of the Figures are poor discussed and interpreted. For example “All data indicated the significant protective effect of CS on PL for proteases and heat treatment. The PL loaded with CS or CS-NCC over 72 h had significantly higher proliferation than those unprotected PL. This result indicates that CS can 614 protect the GF present in PL, thereby promoting fast wound healing.” -this do not say much things. This apply for above Figures too

3.11. CS stabilizing effect on PL” is now improved with more discussion and supported with previous studies (Line- 650).

Discussion is also improved for live/dead assay and cell proliferation assay with references (Line- 614 for live/dead assay) and (Line-582) respectively.

Too many reference out of date and less novel one !

References are replaced with recent ones but those with fundamentals are retained.

These are our best responses to your queries. Thank you very much for your time.

Yours sincerely,

SHIOW- FERN NG

LAW JIA-XIAN

PAYAL BHATNAGAR

24 Nov 2021

Reviewer 2 Report

“to formulate a vehicle” this formulation is out of context !

The abstract is to verbally and requires some brief quantitative results !

The same apply for conclusion !

“Therefore, the use of kenaf-based NCC as wound dressing has yet to be explored.” This seems rather a commercial challenge than a scientific one !

“R & M chemicals, UK “ company was difficult to identify  (please provide a link )

A space between value and units should be, i.e. “4°C” and many other pleas check the entire paper for this problem.

“-80°C” this is not very cold  temperature ?

Equations 2, 5 should be written with professional equation not a photo !

Define all the acronyms before their first appearance in text, i.e. “SEM” and so on

Please provide the approval “This study was approved by the Universiti Kebangsaan

Provide details of “ISO protocol”

What does means this “CTERM”?

How was controlled the humidity “in humidified 5%”?

The statistical analysis is very poor presented, such as difficult to understand the evaluation process

“of PL must be optimum to exert” what does means optimum here?

The Image in Figure 2 are too fuzzy and difficult to evaluate them properly- better quality is required and also the scale bar should be really visible

“the swelling ratio owing to its large hydrophilic groups”- endorse it by a ref!

“bound protein at or near the matrix surface that leached through the membrane”- endorse it by a ref! other wise not sure from where come this assumption !

Figure 9 contain too many Images therefore I suggest splitting it in 3 Figure separately !

Otherwise Figure 9c requires a scale for each image

Figure 9c first raw apart of last image( very down one) the image are the same ! and claiming different composition ! Please check this and all other images from this Figure !

Figure 9 a is very fuzzy and requires scale bar ! the same to many image for a Figure !

Most of the Figures are poor discussed and interpreted. For example “All data indicated the significant protective effect of CS on PL for proteases and heat treatment. The PL loaded with CS or CS-NCC over 72 h had significantly higher proliferation than those unprotected PL. This result indicates that CS can 614 protect the GF present in PL, thereby promoting fast wound healing.” -this do not say much things. This apply for above Figures too

Too many reference out of date and less novel one !

Author Response

(The authors gave the same response as above.)

Round 2

Reviewer 1 Report

The manuscript polymers-1453436 has been improved over the previous version and therefore, I recommend the publication of this work in its present form.

Reviewer 2 Report

-